# Trends in the burden of sickle cell disorders in Sierra Leone, 1990–2023: An analysis of Global Burden of Disease Study 2023 estimates

Monalisa M. J. Faulkner[1,2], Fatima Jalloh[3], Foray Mohamed Foray[4], Sahr L. Gborie[1], Mohamed B. Jalloh[3]*

1 University of Sierra Leone Teaching Hospitals Complex, Ola During Children's Hospital, Freetown, Sierra Leone, 2 Department of Child Health, Korle Bu Teaching Hospital, Accra, Ghana, 3 College of Medicine and Allied Health Sciences, University of Sierra Leone, Freetown, Sierra Leone, 4 College of Health Sciences and Public Policy, Walden University, Minneapolis, Minnesota, United States of America

* bellajay208@gmail.com

## Abstract

### Background

Sickle cell disease is a major cause of childhood mortality in sub-Saharan Africa, yet country-specific burden estimates for high-prevalence settings in West Africa remain limited.

### Objective

To describe Global Burden of Disease (GBD) 2023 modeled estimates of sickle cell disorders burden in Sierra Leone from 1990 through 2023, including temporal trends, age and sex patterns, and demographic contributors to mortality change.

### Methods

We analyzed GBD 2023 modeled estimates for sickle cell disorders in Sierra Leone, including prevalence, deaths, years lived with disability (YLDs), years of life lost (YLLs), and disability-adjusted life-years (DALYs) as absolute counts and age-standardized rates. Temporal trends in age-standardized rates were assessed using log-linear regression. The Kitagawa-Das Gupta decomposition partitioned the change in estimated deaths into population growth, age-structure change, and age-specific mortality-rate changes.

### Results

Estimated prevalent cases increased from 48,689 (95% UI, 42,588−56,140) in 1990–90,498 (78,126−105,815) in 2023. Estimated deaths increased from 408 (288−579) to 635 (438−862), while the estimated age-standardized mortality rate declined from 10.2 to 7.9 per 100,000 (APC, −0.46%; 95% CI, −0.64 to −0.29). Decomposition

**Data availability statement:** The Global Burden of Disease (GBD) 2023 summary estimates analyzed in this study are publicly available from the Institute for Health Metrics and Evaluation (IHME) data portal (GBD Results tool): https://vizhub.healthdata.org/gbd-results/. All analytical code used for data extraction, processing, and analyses is available at: https://github.com/ishmaildiallo/gbd-scd-sierra-leone-1990-2023.

**Funding:** The author(s) received no specific funding for this work.

**Competing interests:** The authors declare no competing interests.

attributed 159.6% of the net increase in deaths to population growth, −7.8% to age-structure change, and −51.8% to lower modeled age-specific rates. In 2023, an estimated 49.5% of deaths occurred before age 20. Point estimates suggested possible higher male mortality, but uncertainty intervals were wide and compatible with no clear sex difference.

## Conclusions

GBD estimates suggest that Sierra Leone's absolute burden of sickle cell disorders increased substantially between 1990 and 2023, while modeled rates declined. These modeled estimates highlight a growing absolute burden and persistent early-life mortality, supporting the need for improved surveillance, newborn screening, infection prophylaxis, hydroxyurea access, and longitudinal care systems.

## Introduction

Sickle cell disorders (SCD) are common, severe inherited hemoglobin disorders and an important cause of premature death and disability worldwide [1–3]. Recent Global Burden of Disease (GBD) estimates showed that babies born with sickle cell disease increased from 453,000 in 2000–515,000 in 2021, while prevalent cases rose from 5.5 million to 7.7 million over the same period [1–3]. The term SCD here denotes the GBD cause category, which includes sickle cell anemia (HbSS), sickle-hemoglobin C disease (HbSC), and sickle-beta-thalassemia (HbS/beta-thalassemia). Sickle cell disease is used when referring more broadly to clinical services, care delivery, and intervention evidence. Sub-Saharan Africa (SSA) bears the greatest burden of sickle cell disease. A recent systematic review and modelling analysis estimated prevalence at 1.54% in infants, 1.51% in children younger than 5 years, and 1.78% in those younger than 15 years, corresponding in 2023 to 1.2 million, 2.8 million, and 8.9 million affected children, respectively [4]. Burden was highest in west and central Africa, with west Africa contributing the largest number of paediatric cases [4].
    [4].
The high burden reflects hemoglobin S carrier frequency, large birth cohorts, malaria ecology, and weak early-life diagnosis. In malaria-endemic SSA, sickle cell trait frequencies commonly range from 10% to 40%, sustaining high inheritance rates [4,5]. Newborn screening remains uncommon, diagnosis is often delayed until severe illness, and comprehensive care is limited [4,6]. Excess mortality is concentrated in early childhood, although survival estimates remain uncertain because population-based cohorts and vital registration data are scarce [7,8].

    Effective interventions are available. Newborn screening, parental education, penicillin prophylaxis, vaccination, malaria prevention, transfusion support, hydroxyurea therapy, and structured follow-up can reduce avaoidable morbidity and early mortality when delivered through reliable health systems [9–11]. Evidence from Jamaica showed that newborn diagnosis linked to organized follow-up improved childhood survival, although these findings demonstrate intervention potential rather than

comparable gains in Sierra Leone [10,11]. More recent African initiatives, including the Consortium on Newborn Screening in Africa, have advanced newborn screening linked to standardized early care as a feasible pathway for high-burden settings [9].

Sierra Leone is thought to carry one of the highest burdens of sickle cell disease in West Africa. Sickle cell trait frequencies have been estimated at 20% to 25%, although nationally representative contemporary data are limited [12,13]. Modeling studies suggest several thousand affected births annually [4,13]. Hospital-based reports indicate that sickle cell disease contributes to severe anemia, hospital admissions, and child mortality, but specialized services, confirmatory diagnostic pathways, and longitudinal follow-up remain limited [12,14–16]. Sierra Leone has no documented national newborn screening program, although point-of-care screening has been piloted at selected sites [15,16].

The evidence gap is therefore both epidemiologic and programmatic. In settings without comprehensive vital registration, newborn screening registries, or longitudinal sickle cell cohorts, primary data are often insufficient for estimating national burden. Country-specific modelled estimates can support policy by providing internally consistent estimates over time, quantifying uncertainty, and making demographic drivers explicit. These estimates should be interpreted as modelled burden estimates rather than observed surveillance data or direct measures of service performance.

The GBD Study provides standardized modelled estimates of mortality, years of life lost (YLLs), years lived with disability (YLDs), and disability-adjusted life-years (DALYs) across countries, causes, years, age groups, and sexes [17]. We used GBD 2023 estimates to describe trends in the burden of sickle cell disorders in Sierra Leone from 1990 through 2023. We characterized age and sex patterns and decomposed the change in estimated deaths into contributions from population growth, population aging, and changes in age-specific mortality rates.

## Methods

### Study design and data source

We conducted a secondary analysis of GBD 2023 modelled estimates produced by the Institute for Health Metrics and Evaluation. The GBD uses a standardized cause hierarchy, systematic data synthesis, and Bayesian meta-regression (DisMod-MR 2.1) to produce internally consistent estimates across 204 countries [17,18]. We extracted all available data for Sierra Leone for sickle cell disorders (GBD cause ID 615) for 1990–2023. No modification of the underlying GBD models was undertaken. Our work represents a country-focused reanalysis of publicly available GBD 2023 modelled estimates [18].

The GBD sickle cell disorders cause category includes sickle cell anemia (HbSS), sickle-hemoglobin C disease (HbSC), and sickle-beta-thalassemia (HbS/beta-thalassemia) as captured within the GBD cause hierarchy. Genotype-specific estimates were not available for this country-level analysis.

### Measures and metrics

We examined five GBD burden measures: prevalence, deaths, YLDs, YLLs, and DALYs. For each, we analyzed absolute counts and age-standardized rates per 100,000 population (computed by GBD using the global reference age structure). All estimates are accompanied by 95% uncertainty intervals (UIs) from 1000 posterior-draw-level simulations [17].

### Age and sex stratification

We summarized modeled burden for non-overlapping age groups: younger than 5 years, 5–19 years, and 20 years and older, in addition to all ages combined. These groupings These groups were selected to distinguish early childhood, school-age and adolescent years, and adulthood [3]. Counts for ages 5–19 years were derived by subtracting estimates for children younger than 5 years from estimates for persons younger than 20 years. Counts for persons aged 20 years and older were derived by subtracting estimates for persons younger than 20 years from all-age estimates. All metrics

were stratified by sex.We additionally present a summary for persons younger than 20 years given the programmatic relevance of this threshold.

### Temporal trend analysis

We fitted log-linear regression models of the form $\ln(rate) = \beta_0 + \beta_1 \times (year)$. The annual percentage change (APC) was calculated as $[\exp(\beta_1) - 1] \times 100$, with 95% confidence intervals from the standard error of $\beta_1$.

### Decomposition of mortality change

We applied the Kitagawa-Das Gupta decomposition to partition the change in deaths between 1990 and 2023 into three additive components: population growth, shifts in age structure, and changes in age-specific mortality rates [19].

Total deaths were expressed as $D = P \times \Sigma(c_i \times r_i)$, where P is total population, $c_i$ is the proportion in age group $i$, and $r_i$ is the age-group–specific death rate. The decomposition first separates total change into a population-size component and a crude-rate component, then separates the crude-rate component into age-structure and age-specific-rate effects. Components were averaged symmetrically across the two time points and sum exactly to the net change in deaths [19,20].

Population denominators were obtained directly from the GBD 2023 population estimates for Sierra Leone, downloaded from the GBD Results Tool by age group (<5 years, 5–14 years, 15–19 years, 20–24 years, and 25+years), sex, and year. These were aggregated into the three decomposition age bands: younger than 5 years (direct), 5–19 years (sum of 5–14 and 15–19), and 20 years and older (sum of 20–24 and 25+). Age-specific mortality rates were then computed directly as deaths divided by population within each group. This approach keeps the decomposition entirely within the GBD estimation framework, using GBD-produced population denominators rather than recovering populations from burden estimates or relying on external sources.

### Sex-based differences

Male-to-female ratios for each burden metric in 2023 were calculated within each age group. Approximate 95% UIs for the ratios were estimated using the delta method, with standard errors derived from the width of the GBD 95% UIs for male and female counts. We note that this approach treats the GBD posterior-derived UIs as though they approximate normal-distribution–based standard errors, which is an approximation; the true posterior distributions may be asymmetric.

### Software

All analyses were conducted in Python 3.12 using pandas (data manipulation), NumPy (numerical operations), SciPy (statistical modelling), and matplotlib (visualization). Analytical code is available at: https://github.com/ishmaildiallo/gbd-scd-sierra-leone-1990-2023.

### Ethical consideration

This study used publicly available, deidentified GBD 2023 estimates. GBD 2023 summary estimates are publicly available through the Institute for Health Metrics and Evaluation data portal. Available from https://vizhub.healthdata.org/gbd-results/. Institutional review board approval was not required.

## Results

### Overall burden and temporal trends

Between 1990 and 2023, the estimated absolute burden of sickle cell disorders in Sierra Leone increased across all measures (Table 1; Fig 1). Estimated prevalent cases rose from 48,689 (95% UI, 42,588−56,140) to 90,498

**Table 1. Estimated burden of sickle cell disorders in Sierra Leone, 1990 and 2023, and percentage change in counts and age-standardized rates.**

| Measure | 1990 Count (95% UI) | 1990 ASR per 100,000 | 2023 Count (95% UI) | 2023 ASR per 100,000 | Count Change, % | Rate Change, % |
|---|---|---|---|---|---|---|
| Prevalence | 48,689 (42,588–56,140) | 1215.0 (1062.7–1400.9) | 90,498 (78,126–105,815) | 1129.3 (974.9–1320.4) | +85.9% | −7.1% |
| Deaths | 408 (288–579) | 10.2 (7.2–14.4) | 635 (438–862) | 7.9 (5.5–10.8) | +55.7% | −22.1% |
| YLDs | 3,634 (2,462–5,217) | 90.7 (61.4–130.2) | 6,843 (4,605–9,286) | 85.4 (57.5–115.9) | +88.3% | −5.8% |
| YLLs | 28,884 (20,422–41,472) | 720.8 (509.6–1034.9) | 43,765 (29,905–60,483) | 546.1 (373.2–754.7) | +51.5% | −24.2% |
| DALYs | 32,518 (23,446–45,336) | 811.5 (585.1–1131.3) | 50,608 (36,331–66,317) | 631.5 (453.4–827.5) | +55.6% | −22.2% |

*Abbreviations: ASR, age-standardized rate; CI, confidence interval; DALYs, disability-adjusted life-years; UI, uncertainty interval; YLDs, years lived with disability; YLLs, years of life lost. Count change and rate change are expressed as percentage change from 1990 to 2023. Rates are age-standardized per 100,000 population.*

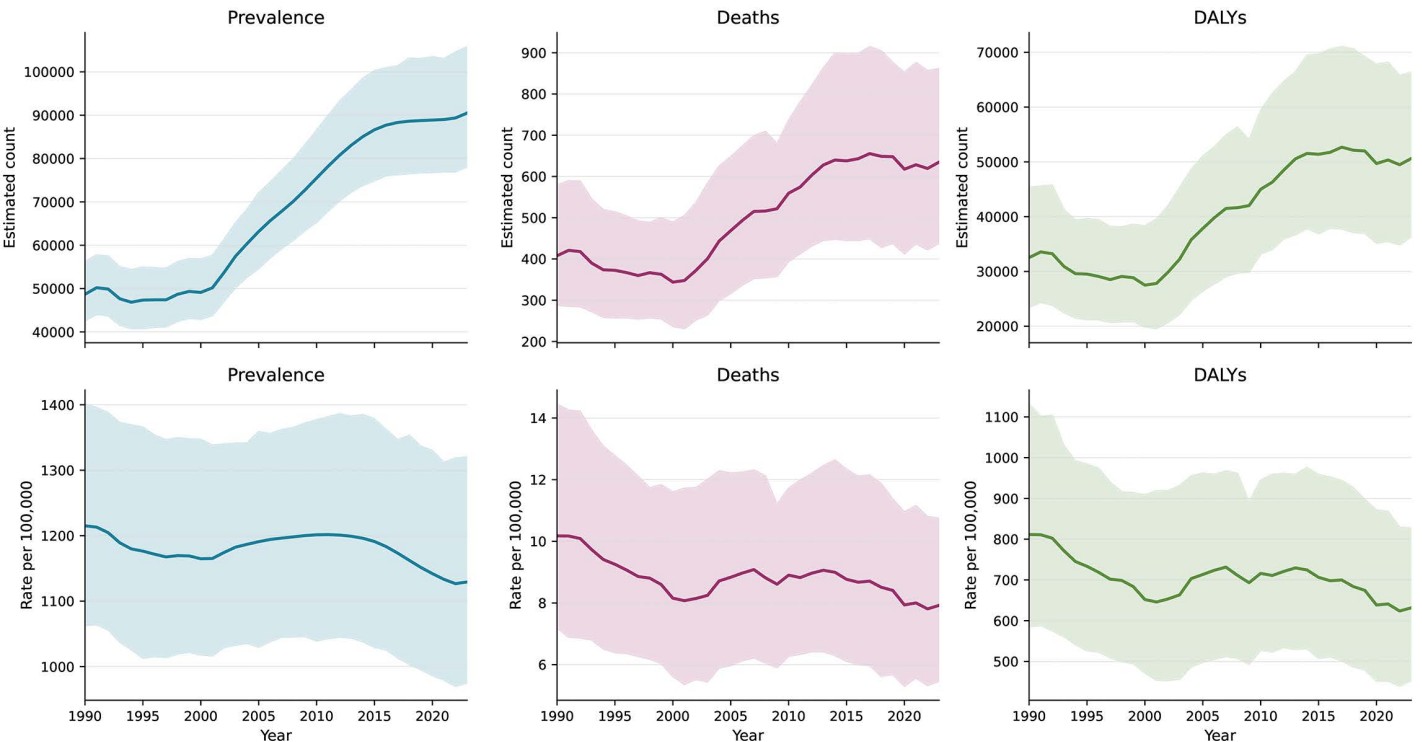

**Fig 1. Temporal trends in estimated sickle cell disorders burden in Sierra Leone, 1990-2023.** Top: absolute estimated counts. Bottom: estimated age-standardized rates per 100,000 population. Shaded bands, 95% UIs.

(78,126−105,815), an increase of 85.9%. Estimated deaths increased from 408 (288−579) to 635 (438−862; +55.7%), and DALYs from 32,518 (23,446−45,336) to 50,608 (36,331−66,317; +55.6%).

Despite rising absolute counts, estimated age-standardized rates declined. The age-standardized mortality rate decreased from 10.2 (7.2–14.4) to 7.9 (5.5–10.8) per 100,000 (−22.1%), and the age-standardized DALY rate from 811.5 (585.1–1131.3) to 631.5 (453.4–827.5) per 100,000 (−22.2%). Log-linear regression confirmed statistically significant

estimated annual declines for mortality (APC, −0.46%; 95% CI, −0.64 to −0.29; P<.0001), DALYs (APC, −0.43%; 95% CI, −0.60 to −0.25; P=.0001), and YLLs (APC, −0.48%; 95% CI, −0.68 to −0.29; P<.0001). The prevalence rate declined modestly (APC, −0.10%; P=.003), while the YLD rate showed no significant trend (APC, −0.01%; P=.82) (Table 2).

### Age distribution of estimated burden in 2023

In 2023, an estimated 153 deaths (95% UI, 93–230; 24.1% of total) occurred among children younger than 5 years, 162 deaths (25.4%) among persons aged 5–19 years, and 320 deaths (50.5%) among adults aged 20 years and older (Table 3; Fig 2). Taken together, persons younger than 20 years accounted for approximately half (49.5%) of all estimated deaths. Prevalence was highest among persons aged 5–19 years (42,580 cases; 47.1%), followed by children younger than 5 (26,189; 28.9%) and adults aged 20 and older (21,729; 24.0%). DALYs were distributed more evenly, with the 20-and-older group contributing the largest share (20,171; 39.9%), followed by 5–19 years (15,814; 31.2%) and younger than 5 years (14,623; 28.9%).

### Sex-based differences

Males and females had near-equal estimated prevalence (45,442 vs 45,056 cases; male-to-female ratio, 1.01; 95% UI, 0.78–1.23). However, males had higher estimated mortality: 371 versus 264 deaths (ratio, 1.40; 95% UI, 0.47–2.33), with similar patterns for DALYs (ratio, 1.33; 95% UI, 0.53–2.12) and YLLs (ratio, 1.40; 95% UI, 0.45–2.35). YLDs showed a slight female preponderance (ratio, 0.94; 95% UI, 0.45–1.43). These point estimates suggest a pattern of excess male mortality despite similar disease frequency, although the wide UIs preclude firm conclusions (Fig 2; Fig 4C).

### Decomposition of mortality change

The Kitagawa-Das Gupta decomposition attributed the net increase of 227 estimated deaths to three countervailing forces (Fig 3). Population growth was the dominant driver, contributing +363 additional deaths (+159.6% of the net change); Sierra Leone's GBD-estimated population approximately doubled during the study period, from 4.0 million to 8.0 million. Age-structure shifts contributed −18 deaths (−7.8%), consistent with a declining proportion of children younger than 5 years (from 18.0% to 13.7%) and growing proportion of adults aged 20 years and older (from 48.4% to 52.5%). Declining age-specific mortality rates offset a substantial portion of the population-driven increase (−118 deaths; −51.8%), indicating that, in the absence of population growth and demographic shifts, sickle cell mortality would have decreased. The three components summed exactly to the total observed change Fig 4.

**Table 2. Estimated annual percentage change in age-standardized rates of sickle cell disorders burden in Sierra Leone, 1990-2023, from log-linear regression of GBD 2023 modelled rates.**

| Measure | APC, % (95% CI) | P Value | R² |
|---|---|---|---|
| Prevalence | −0.10 (−0.16 to −0.04) | 0.003 | 0.243 |
| Deaths | −0.46 (−0.64 to −0.29) | < 0.001 | 0.454 |
| YLDs | −0.01 (−0.08 to 0.07) | 0.82 | 0.002 |
| YLLs | −0.48 (−0.68 to −0.29) | < 0.001 | 0.419 |
| DALYs | −0.43 (−0.60 to −0.25) | < 0.001 | 0.401 |

*Abbreviations: APC, annual percentage change; CI, confidence interval; DALYs, disability-adjusted life-years; YLDs, years lived with disability; YLLs, years of life lost. APC was estimated by log-linear regression of the age-standardized rate on year. A negative APC indicates a declining rate over the study period.*

**Table 3. Estimated age and sex distribution of sickle cell disorders burden in Sierra Leone, 2023, using non-overlapping age groups. The 5-to-19-year and 20-and-older groups were derived by subtraction (see Methods). Male-to-female ratios with approximate 95% UIs estimated by the delta method.**

| Measure | Age Group | Sex | Estimate | Lower UI | Upper UI | % of Total |
|---|---|---|---|---|---|---|
| Prevalence | <5 years | Both | 26,189 | 22,349 | 29,463 | 28.9 |
| | <5 years | Male | 12,863 | 10,877 | 14,630 | 14.2 |
| | <5 years | Female | 13,326 | 11,245 | 15,235 | 14.7 |
| | 5-19 years | Both | 42,580 | 30,065 | 57,454 | 47.1 |
| | 5-19 years | Male | 21,536 | 15,159 | 29,727 | 23.8 |
| | 5-19 years | Female | 21,044 | 14,180 | 28,588 | 23.3 |
| | 20+years | Both | 21,729 | 0 | 46,287 | 24.0 |
| | 20+years | Male | 11,042 | 0 | 23,673 | 12.2 |
| | 20+years | Female | 10,687 | 0 | 23,313 | 11.8 |
| | All ages | Both | 90,498 | 78,126 | 105,815 | 100.0 |
| | All ages | Male | 45,442 | 39,099 | 53,462 | 50.2 |
| | All ages | Female | 45,056 | 38,510 | 52,728 | 49.8 |
| Deaths | <5 years | Both | 152.9 | 92.7 | 230.2 | 24.1 |
| | <5 years | Male | 86.4 | 44.4 | 147.3 | 13.6 |
| | <5 years | Female | 66.5 | 38.4 | 108.4 | 10.5 |
| | 5-19 years | Both | 161.6 | 0.0 | 353.6 | 25.4 |
| | 5-19 years | Male | 101.9 | 0.0 | 264.6 | 16.0 |
| | 5-19 years | Female | 59.7 | 0.0 | 156.8 | 9.4 |
| | 20+years | Both | 320.4 | 0.0 | 658.4 | 50.5 |
| | 20+years | Male | 182.5 | 0.0 | 498.9 | 28.7 |
| | 20+years | Female | 138.0 | 0.0 | 304.4 | 21.7 |
| | All ages | Both | 634.9 | 438.3 | 861.5 | 100.0 |
| | All ages | Male | 370.7 | 204.0 | 596.3 | 58.4 |
| | All ages | Female | 264.2 | 173.5 | 384.5 | 41.6 |
| DALYs | <5 years | Both | 14622.7 | 9127.4 | 21750.0 | 28.9 |
| | <5 years | Male | 8220.3 | 4417.4 | 13506.3 | 16.2 |
| | <5 years | Female | 6402.4 | 3786.2 | 10013.8 | 12.7 |
| | 5-19 years | Both | 15814.5 | 0.0 | 31640.2 | 31.2 |
| | 5-19 years | Male | 9437.1 | 0.0 | 22681.0 | 18.6 |
| | 5-19 years | Female | 6377.4 | 0.0 | 14430.9 | 12.6 |
| | 20+years | Both | 20170.7 | 0.0 | 45013.9 | 39.9 |
| | 20+years | Male | 11197.7 | 0.0 | 34784.1 | 22.1 |
| | 20+years | Female | 8973.0 | 0.0 | 21687.3 | 17.7 |
| | All ages | Both | 50607.9 | 36330.8 | 66316.5 | 100.0 |
| | All ages | Male | 28855.1 | 16601.9 | 44762.5 | 57.0 |
| | All ages | Female | 21752.8 | 15128.3 | 30325.5 | 43.0 |

*Abbreviations: DALYs, disability-adjusted life-years; UI, uncertainty interval. Age groups are non-overlapping: < 5 years, 5–19 years, 20+years, and all ages. Percentage of total is calculated relative to the all-ages, both-sexes estimate for each measure.*

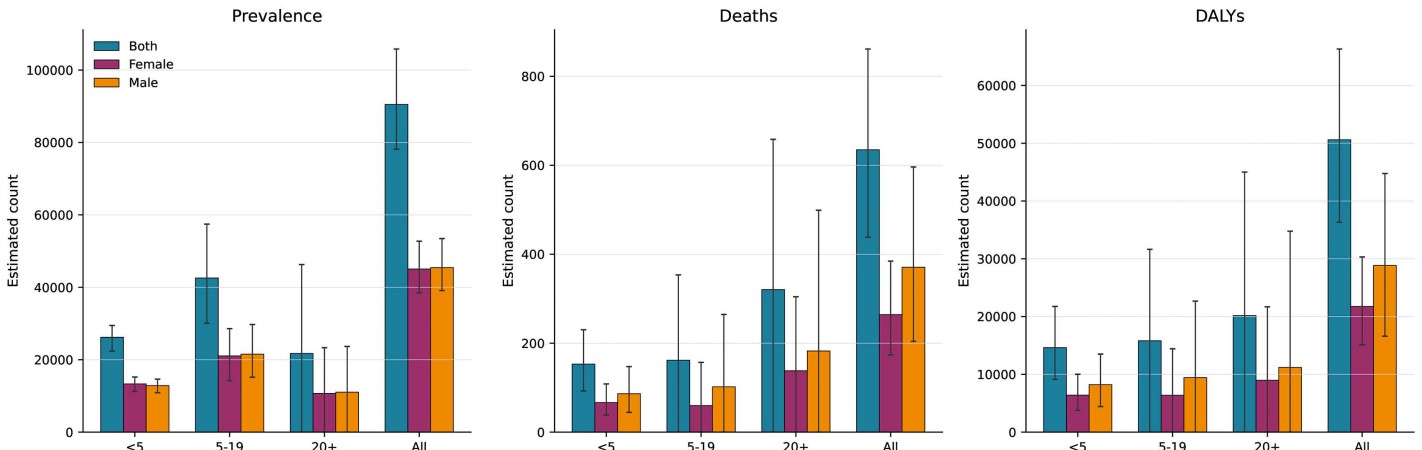

**Fig 2. Estimated age and sex distribution of sickle cell disorders burden in Sierra Leone, 2023. (A)** Prevalence, **(B)** deaths, and **(C)** DALYs by non-overlapping age group and sex. Error bars, 95% UIs for direct GBD estimates; derived groups carry additional uncertainty.

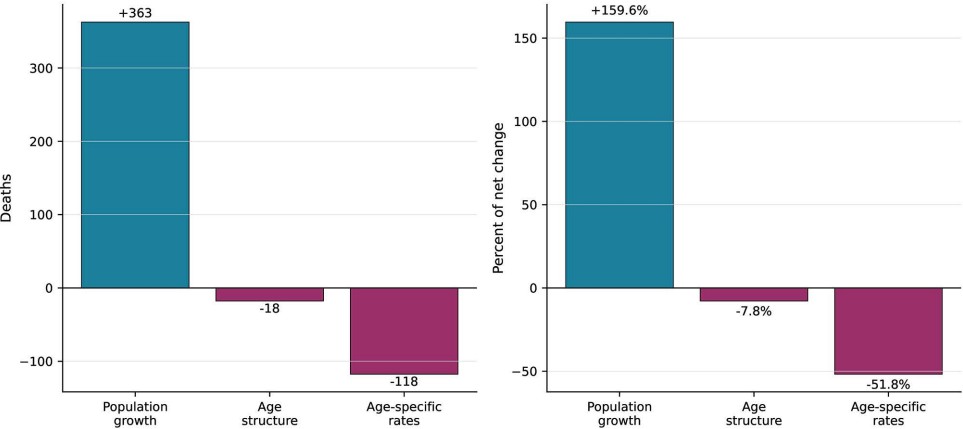

**Fig 3. Kitagawa-Das Gupta decomposition of the change in estimated sickle cell disorders deaths, Sierra Leone, 1990−2023.** The net increase of 227 estimated deaths was partitioned into population growth (+363; +159.6%), age-structure change (−18; −7.8%), and lower modeled age-specific mortality rates (−118; −51.8%). Components sum to the total change.

## Discussion

In this analysis of GBD 2023 modelled estimates spanning more than three decades, we found that the estimated absolute burden of sickle cell disorders in Sierra Leone increased substantially, while estimated age-standardized mortality and DALY rates declined. Decomposition indicated that population growth accounted for most of the increase in estimated deaths, whereas lower modelled age-specific mortality rates partly offset population-related increases. Estimated mortality remained concentrated in children and adolescents, with nearly half of deaths occurring before age 20. Point estimates suggested possible sex differences in mortality, but uncertainty was substantial. These findings should therefore be viewed as hypothesis-generating.

These findings are consistent with broader global and regional evidence. Recent GBD analyses showed that the number of babies born with sickle cell disease increased from 453,000 in 2000–515,000 in 2021, while prevalent cases

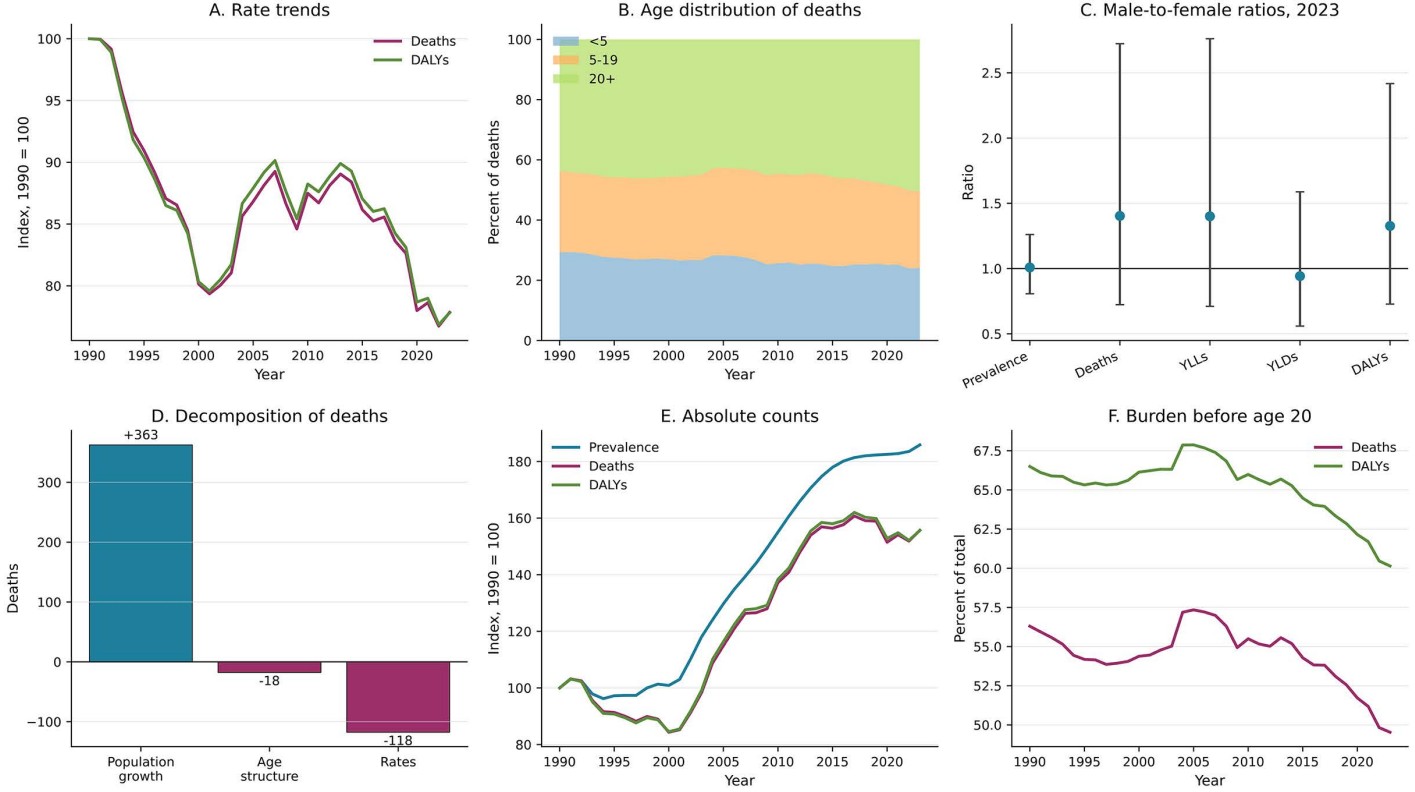

**Fig 4. Summary of modeled sickle cell disorders burden estimates.** (A) Indexed mortality and DALY rate trends. (B) Proportional age distribution of estimated deaths. (C) Male-to-female ratios with approximate 95% uncertainty intervals. (D) Decomposition components. (E) Indexed absolute count trends. (F) Proportion of estimated deaths and DALYs in persons younger than 20 years.

increased from 5.5 million to 7.7 million over the same period [1,2]. This pattern indicates that absolute burden can rise even when age-standardized rates decline. It is particularly relevant for West African countries, where high hemoglobin S carrier frequencies, large birth cohorts, and weak early-life diagnostic systems intersect.

Recent paediatric estimates from SSA place our findings in context. A systematic review and modelling analysis estimated sickle cell disease prevalence at 1.54% in infants, 1.51% in children younger than 5 years, and 1.78% in those younger than 15 years, corresponding to 1.2 million, 2.8 million, and 8.9 million affected children, respectively, in 2023 [4]. However, substantial heterogeneity and moderate study quality limit precision, especially in countries with few population-based data.

Our estimates should therefore be interpreted as modelled burden estimates, not as surveillance data or measures of service performance. Declines in age-standardized mortality and DALY rates may reflect model structure, background mortality, covariate patterns, regional borrowing, or demographic change. They should not be read as evidence of improved sickle cell care. Where screening, vital registration, diagnostic confirmation, and follow-up are limited, modelled trends can support planning but cannot establish survival gains.

The findings also align with regional evidence that the early-life burden is substantial and incompletely measured. Across much of SSA, newborn screening remains uncommon, diagnosis is often delayed until severe illness, and comprehensive care remains limited [4,21]. Mortality estimates in children with sickle cell disease are especially uncertain because many deaths occur before diagnosis and may be assigned to malaria, sepsis, severe anemia, malnutrition, or

other proximate causes. These competing classifications are not just coding issues. They shape whether sickle cell disease is visible to families, clinicians, and ministries of health.

Estimates derived from cross-sectional prevalence data do not necessarily reflect survival trajectories. Older children may be more likely to survive long enough to receive a diagnosis, whereas infants may die before ascertainment where newborn screening is unavailable [4]. Age-specific burden patterns in Sierra Leone may therefore reflect disease occurrence, survival, diagnostic access, and modelled mortality assumptions. Stronger primary data are needed to distinguish these components. The policy implications of these findings must be read against the reality of Sierra Leone's current health system. Sierra Leone had no national newborn screening program for sickle cell disease as of 2024, though point-of-care screening had been piloted at Ola During Children's Hospital in Freetown and at rural sites through the EASEL project [15,16]. Prevalent cases nearly doubled over 33 years, yet no systematic identification pathway exists. A national or sentinel registry linking screening, genotype, clinical encounters, treatment exposure, complications, and survival would enable comparison of modelled estimates with observed outcomes and identify gaps between diagnosis and care enrolment. Embedding sickle cell testing within malaria, HIV, or other population surveys linked to district health information systems would strengthen primary data collection [4].

Point-of-care testing integrated into maternity discharge and immunization contacts could reach this group, but screening without confirmatory testing, counselling, referral, and treatment access has limited value. Early care should include penicillin prophylaxis, folic acid, malaria prevention, routine vaccination, caregiver education, and prompt evaluation for fever, severe anaemia, splenic enlargement, and respiratory symptoms [3,22,23].

Age-specific mortality rates declined over the study period despite the absence of formal sickle cell infrastructure, suggesting some improvement in survival. Sustaining that trajectory would require moving beyond episodic crisis management. Hydroxyurea remains absent from Sierra Leone's essential medicines list and accessible mainly through donor-funded initiatives [24,25]. Scale-up would require national listing, simplified dosing protocols, laboratory monitoring, pharmacovigilance, and workforce readiness, alongside a national sickle cell centre with a registry linked to newborn screening [11,26,27].

Our estimates indicated a possible male excess in estimated mortality (ratio, 1.40; 95% UI, 0.47–2.33), but the uncertainty interval is wide, includes unity, and does not exclude the possibility of no difference or a reversal. Whether this reflects differences in care-seeking, complication rates, or ascertainment cannot be determined from modelled data alone and would require longitudinal registry data to investigate.

The 2014−2015 Ebola epidemic was associated with reduced maternal and child-health service use and an estimated 20% increase in national under-5 mortality [28,29]. Because persons under 20 account for nearly half of estimated sickle cell deaths in this analysis, service continuity during health-system disruptions is a relevant planning consideration. Incorporating decentralized drug supply, community follow-up, and continuity mechanisms into sickle cell programme design may reduce vulnerability to outbreaks and workforce shortages.

A content analysis of 804 sickle cell articles across 13 African countries from 2006 to 2023 found Sierra Leone contributed only three, with newborn screening covered in 20.8% of articles regionally [30]. The burden documented here will not translate into care-seeking or policy action without communication strategies that reframe sickle cell disease as a treatable chronic condition.

## Strengths and limitations

This study has several strengths. It provides a country-specific analysis for Sierra Leone, a high-priority West African setting with limited population-level sickle cell data. It uses a standardized GBD 2023 framework across more than three decades, reports uncertainty intervals, summarizes age and sex patterns, and applies decomposition to distinguish population growth, age-structure change, and age-specific mortality-rate change.

The study is not without limitations. First, all results are based on GBD modelled estimates. Primary data from Sierra Leone are sparse, and estimates may depend on covariates, related causes, and regional borrowing of information. Second, early childhood deaths may be underrepresented when children die before diagnosis or when deaths are assigned to infections, anemia, malaria, or malnutrition. Third, rate declines cannot be interpreted causally. They may reflect model assumptions, demographic shifts, background mortality patterns, or changes unrelated to sickle cell-specific care. Fourth, the GBD sickle cell disorders category does not permit genotype-specific analysis for HbSS, HbSC, and HbS/beta-thalassemia in Sierra Leone. Finally, derived non-overlapping age groups and approximate sex ratios may add uncertainty beyond the reported intervals.

## Conclusion

GBD 2023 modeled estimates suggest that the absolute burden of SCDs in Sierra Leone increased substantially between 1990 and 2023, while age-standardized mortality and DALY rates declined. Population growth accounted for most of the increase in estimated deaths, partly offset by lower modelled age-specific mortality rates. These results should be interpreted cautiously because surveillance is limited. They nevertheless identify a growing need for primary data, newborn and early-childhood screening, infection prophylaxis, hydroxyurea access, longitudinal care, and public engagement around sickle cell disease in Sierra Leone.

## Supporting information

**S1 File. Supporting information.**
(DOCX)

## Acknowledgments

The authors acknowledge the Institute for Health Metrics and Evaluation and GBD collaborators for producing and sharing the 2023 estimates, including the population denominators used in this analysis.

## Author contributions

**Conceptualization:** Mohamed B. Jalloh.

**Data curation:** Monalisa M.J. Faulkner, Mohamed B. Jalloh.

**Formal analysis:** Monalisa M.J. Faulkner, Mohamed B. Jalloh.

**Investigation:** Monalisa M.J. Faulkner, Fatima Jalloh, Foray Mohamed Foray, Sahr L. Gborie, Mohamed B. Jalloh.

**Methodology:** Monalisa M.J. Faulkner, Mohamed B. Jalloh.

**Project administration:** Monalisa M.J. Faulkner, Foray Mohamed Foray, Mohamed B. Jalloh.

**Resources:** Monalisa M.J. Faulkner, Fatima Jalloh, Foray Mohamed Foray, Sahr L. Gborie, Mohamed B. Jalloh.

**Software:** Mohamed B. Jalloh.

**Supervision:** Mohamed B. Jalloh.

**Validation:** Monalisa M.J. Faulkner, Fatima Jalloh, Foray Mohamed Foray, Sahr L. Gborie, Mohamed B. Jalloh.

**Visualization:** Mohamed B. Jalloh.

**Writing – original draft:** Monalisa M.J. Faulkner, Mohamed B. Jalloh.

**Writing – review & editing:** Monalisa M.J. Faulkner, Fatima Jalloh, Foray Mohamed Foray, Sahr L. Gborie, Mohamed B. Jalloh.

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
