## [Decision Letter · Decision Letter 0]

28 Apr 2026

PONE-D-26-09105Trends in the burden of sickle cell disorders in Sierra Leone, 1990–2023: an analysis of Global Burden of Disease Study 2023 estimatesPLOS One

Dear Dr. Jalloh,

Thank you for submitting your manuscript to PLOS ONE. After careful consideration, we feel that it has merit but does not fully meet PLOS ONE’s publication criteria as it currently stands. Therefore, we invite you to submit a revised version of the manuscript that addresses the points raised during the review process.

**This manuscript yields valuable findings, but it needs to address its reliance on modeled data, tone down some claims, restructure sections, insert the missing figures, and improve the clarity of the English.**

We look forward to receiving your revised manuscript.

Kind regards,

Mehmet Baysal

Academic Editor

PLOS One

**Journal Requirements:**

1. When submitting your revision, we need you to address these additional requirements. Please ensure that your manuscript meets PLOS ONE's style requirements, including those for file naming. The PLOS ONE style templates can be found at https://journals.plos.org/plosone/s/file?id=wjVg/PLOSOne_formatting_sample_main_body.pdf and https://journals.plos.org/plosone/s/file?id=ba62/PLOSOne_formatting_sample_title_authors_affiliations.pdf 2. Your ethics statement should only appear in the Methods section of your manuscript. If your ethics statement is written in any section besides the Methods, please delete it from any other section. 3. Please upload a new copy of Figures 1, 2 and 4, as the detail is not clear. Please follow the link for more information:  https://journals.plos.org/plosone/s/figures 4. Please include captions for your Supporting Information files at the end of your manuscript, and update any in-text citations to match accordingly. Please see our Supporting Information guidelines for more information: http://journals.plos.org/plosone/s/supporting-information. 5. If the reviewer comments include a recommendation to cite specific previously published works, please review and evaluate these publications to determine whether they are relevant and should be cited. There is no requirement to cite these works unless the editor has indicated otherwise.

**Additional Editor Comments:**

This manuscript yields valuable findings, but it needs to address its reliance on modeled data, tone down some claims, restructure sections, insert the missing figures, and improve the clarity of the English.

Reviewers' comments:

Reviewer's Responses to Questions

**Comments to the Author**

1. Is the manuscript technically sound, and do the data support the conclusions?

Reviewer #1: Yes

Reviewer #2: Yes

Reviewer #3: Partly

Reviewer #4: Yes

Reviewer #5: Partly

Reviewer #6: Yes

2. Has the statistical analysis been performed appropriately and rigorously?

Reviewer #1: I Don't Know

Reviewer #2: Yes

Reviewer #3: Yes

Reviewer #4: Yes

Reviewer #5: I Don't Know

Reviewer #6: Yes

3. Have the authors made all data underlying the findings in their manuscript fully available?

Reviewer #1: Yes

Reviewer #2: Yes

Reviewer #3: Yes

Reviewer #4: Yes

Reviewer #5: Yes

Reviewer #6: Yes

4. Is the manuscript presented in an intelligible fashion and written in standard English?

Reviewer #1: Yes

Reviewer #2: Yes

Reviewer #3: Yes

Reviewer #4: Yes

Reviewer #5: No

Reviewer #6: Yes

5. Review Comments to the Author

**Reviewer #1:** Review “Trends in the burden of sickle cell disorders in Sierra Leone, 1990–2023: an

analysis of Global Burden of Disease Study 2023 estimates”

This manuscript examines SCD trends in Sierra Leone from 1990 to 2023 using GBD 2023 data. Findings show that while absolute cases have risen due to population growth, age-standardized rates have declined. Decomposition methods identify factors driving mortality changes. The topic is significant and understudied in high-burden West African regions.

Concerns:

1. Significant Dependence on Modeled Data (Key Limitation)

The manuscript uses only GBD modeled estimates and not primary or surveillance data.

Recommendation: This limitation should be clearly noted in:

• Abstract (currently insufficient emphasis)

• Discussion (needs deeper analysis), including:

• Data sparsity in Sierra Leone

• Reliance on related variables and regional borrowing in GBD models

• Potential circularity of policy conclusions based on modeled instead of observed data

2. The manuscript views lower age-standardized mortality as improvement, but this may result from GBD model assumptions, shifts in background mortality or demographics, rather than actual advances in SCD care.

Recommendation: Use cautious phrasing like “consistent with potential improvements” without implying causality.

3. Sex Differences overinterpretation--The manuscript claims excess male mortality, but the wide UI (0.47–2.33) includes the possibility of no effect or the opposite.

The conclusion lacks statistical support.

Recommendation: Tone down language- Replace “suggested excess male mortality” with: “point estimates indicate possible differences, but uncertainty is substantial”

4. Limited Engagement with Existing Literature--The introduction is strong, but the discussion has inadequate context:

Missing comparison with other West African countries, previous GBD SCD studies, benchmarking against Nigeria, Ghana, etc.

Recommendation:

Include regional comparisons and address how Sierra Leone's trends relate to others—whether typical or unique.

**Reviewer #2:** 1. Is the manuscript technically sound, and do the data support the conclusions?

The authors produced a sound document with data from the Global Burden of Disease (GBD) 2023 that was pre-existing and available in the public domain. All analytical codes utilised for data extraction, and analyses was readily available. The data provided supports the conclusions that were drawn.

2. Has the statistical analysis been performed appropriately and rigorously?

The statistical analyses, though a secondary analyses, was appropriate for the nature of information that the authors wanted to derive. The analysis rightfully encompasses the sickle cell haemoglobinopathies that carry significant morbidity and mortality that include sickle cell anemia (HbSS), sickle-hemoglobin C disease (HbSC), and sickle-beta-thalassemia (HbS/β-thal). Absolute counts of prevalence, deaths, years lived with disability (YLDs), years of life lost (YLLs), and disability-adjusted life-years (DALYs) were recorded. The estimated annual percentage change in age-standardized rates was documented with p-values and confidence intervals. Age and sex stratification was reasonable, and temporal trends were appropriately assessed using log-linear regression. The Kitagawa-Das Gupta decomposition was also an appropriate method to compare mortality trends. The limitations of assumptions made were highlighted in the discussion and some sections of the methodology.

Tables and diagrams were formatted and labelled appropriately.

3. Have the authors made all data underlying the findings in their manuscript fully available?

The authors have made all data, codes utilised for data extraction and analyses readily available on an online platform - GitHub.

4. Is the manuscript presented in an intelligible fashion and written in standard English?

Yes, mostly. Minor corrections highlighted below:

Introduction

The authors describe the elements of the introduction relatively well. The background knowledge of Sickle cell Disease was appropriately mentioned, followed by the current knowledge and the gaps in knowledge. The objectives of this study were also highlighted clearly. The authors have not quite elucidated clearly the rationale for partaking in this study. What will be the impact of this intervention? That is not apparent in the introduction. Is it to change policy for instance?

Methodology

The methodology section should present a clear and factual description of the methods used in the study. Discussion of the limitations of these methods is not appropriate in this section and should instead be addressed in the discussion section.

Conclusions

The authors may need to consider that there is more that may contribute to the ‘health shocks.’ Consider the additive effect of the HIV epidemic, malnutrition, the effect of malaria as well as the effect of the covid 19 outbreak on infant mortality rate in that region during that period.

**Reviewer #3:** Very good paper. The following changes need to be made:

1) abstract section: Objectives of the study need to be clearly stipulated. Separate results from conclusions. Let the conclusion speak to the title, objectives and the results derived from the study which was global burden of disease trends

2) Give the global trends of SCD, then regional then local in the introduction section.

3)Which data collection methods were used in 1990 and how well kept were they? Was the collection standardised back then or can this then be mentioned as a limitation of the study being that it was historical data?

4)In 1990 what tests were being used to diagnose Sickle Cell Disease for confirmatory diagnosis? This needs to be mentioned and what tests are being used now for the same? Were they screening tests or diagnostic tests?

5)Pages 10,11,12 insert the actual figures indicated

6)In the discussion , please compare similar studies done for these trends that may have employed the Kitagawa-Das Gupta Decomposition to explain the findings. Refrain from making recommendations in the discussion section, these can be done after the conclusion

7)Separate the results from the discussion, refrain from explaining the results in the results section, this should be done in the discussion section with relevant studies referenced.

8)The limitations of the study need to be in their own section

9)Conclusion is not clear and conscise. Please make it better in keeping with the title and objectives of the study

10)Let the information flow from one section to the other in keeping with the study title and don't forget to add the objectives as well.

**Reviewer #4:** Overall this is a well-analyzed paper that presents important population-based information regarding SCD in Sierra Leone. Only minor revisions are suggested, including ensuring that the figures are inserted in the appropriate spaces in the paper. Additionally, I would suggest re-examining the paper to see how certain sections could be made more concise. The discussion is quite lengthy and includes speculative statements that may be reduced or consolidated to streamline the paper more.

**Reviewer #5:** Intheir manuscript Jalloh et al elegantly reuse data from a previous staudy to provide in depth analysis od sickle cell disease burder in Sierra leone. Such data are rare and deserve publication, however numerous technical and formal points need to be adresses prior to improve the overall value of the manuscript, here are some examples:

- the results paragraph of the abstract does not protray clearly the content of the paper.

- prevalence is in %, here the authors present absolute value. Personally i encourage to express the data as % of the population to improve readibility.

"sickle cell disaese shoud be defined at the beguining of the manuscript: SS only SC other componds or phenotypically defines?

- the sex base difference is poorly explained and deserved a bit more effort.

In addition to overall statistics should be doubleproofed as well as the English scientific language reviewed.

**Reviewer #6:** I would like to thank the authors for the opportunity to review this manuscript, which describes trends in the burden of sickle cell disease (SCD) in Sierra Leone using a country-specific analysis of the Global Burden of Disease 2023. As the authors highlight in the introduction, this study addresses a critical public health need: while SCD is known for its high prevalence, mortality, and morbidity, precise data to guide policy decisions remain scarce in Sierra Leone.

The manuscript is well written, with a clear justification for the study. The methodology is thoroughly described, and the results are presented with clarity. The discussion effectively focuses on the study’s key finding—the contrast between higher absolute counts across all studied criteria and lower age-standardized rates during the study period. The authors provide a thoughtful analysis of this seemingly paradoxical result, supported by a decomposition of mortality changes. The limitations are also well addressed.

This study will undoubtedly contribute to shaping public health policies aimed at improving SCD management and outcomes in Sierra Leone.

Minor suggestions for revision:

The figures appear blurry and would benefit from higher resolution before publication.

Placeholder text such as “Insert FigXX here” should be removed.

Beyond these minor comments, the article is well suited for publication.

6. PLOS authors have the option to publish the peer review history of their article (what does this mean?). If published, this will include your full peer review and any attached files.

**Do you want your identity to be public for this peer review?** For information about this choice, including consent withdrawal, please see our Privacy Policy.

Reviewer #1: No

Reviewer #2: No

Reviewer #3: **Yes:** AYAYA JOY MUYONGA

Reviewer #4: No

Reviewer #5: No

Reviewer #6: No

---

## [Author Response · Author response to Decision Letter 1]

4 May 2026

May 3, 2026

Emily Chenette, PhD

Editor-in-Chief

PLOS ONE

Dear Dr. Chenette,

Re: Response to reviewers: “Trends in the burden of sickle cell disorders in Sierra Leone, 1990–2023: an analysis of Global Burden of Disease Study 2023 estimates”

We thank the Academic Editor and reviewers for their careful evaluation of the manuscript. We have revised the manuscript to emphasize that all results are GBD 2023 modeled estimates, to reduce causal language, to separate Results from Discussion, to create a distinct limitations discussion, to remove figure placeholders, and to improve clarity and concision.

Academic Editor

Comment: Reliance on modeled data and tone of claims

Response: We revised the Abstract, Introduction, Results, Discussion, Limitations, and Conclusion to state that the estimates are modeled GBD 2023 estimates rather than observed surveillance data. We removed language implying proven improvements in Sierra Leone's sickle cell care. Revised text states: "these GBD estimates should be interpreted as modeled burden estimates rather than direct measures of service performance" and "these trends should be interpreted cautiously".

Comment: Section structure

Response: We restructured the manuscript to separate descriptive Results from interpretation and added Discussion subheadings: Principal findings; Interpretation in global and regional context; Implications for Sierra Leone; Strengths and limitations; and Conclusion.

Comment: Figures

Response: We apologize for the confusion caused by the figure placeholders. In the original submission, high-resolution figures were uploaded separately according to journal instructions, but placeholder text remained in the manuscript file. We have now removed all placeholder text, ensured that each figure is cited in the correct location, retained complete captions at the appropriate points in the manuscript, and uploaded high-resolution versions of Figures 1, 2, 3, and 4 as separate figure files in tiff formats. We also revised Figures 1, 2, and 4 to improve clarity and readability.

Comment: English clarity

Response: We edited the full manuscript for concise scientific prose, consistent use of "modeled estimates," and cautious interpretation throughout the manuscript.

Comment: Ethics statement

Response: We retained the ethics statement only in the Methods section and removed the duplicate ethics declaration from the declarations section. The Data Availability Statement no longer repeats the institutional review board statement.

REVIEWER #1

Comment: Significant dependence on modeled data

This manuscript examines SCD trends in Sierra Leone from 1990 to 2023 using GBD 2023 data. Findings show that while absolute cases have risen due to population growth, age-standardized rates have declined. Decomposition methods identify factors driving mortality changes. The topic is significant and understudied in high-burden West African regions.

Concerns:

1. Significant Dependence on Modeled Data (Key Limitation)

The manuscript uses only GBD modeled estimates and not primary or surveillance data.

Recommendation: This limitation should be clearly noted in:

• Abstract (currently insufficient emphasis)

• Discussion (needs deeper analysis), including:

• Data sparsity in Sierra Leone

• Reliance on related variables and regional borrowing in GBD models

• Potential circularity of policy conclusions based on modeled instead of observed data.

Response: Thank you for emphasizing this key limitation. We strengthened the Abstract, Introduction, and Discussion to state that the analysis uses modeled GBD estimates, not primary surveillance data. We also expanded the limitations to discuss sparse primary data, possible regional borrowing, underascertainment of early deaths, and the need to treat policy implications cautiously.

Comment: Lower rates may not indicate improved SCD care.

2. The manuscript views lower age-standardized mortality as improvement, but this may result from GBD model assumptions, shifts in background mortality or demographics, rather than actual advances in SCD care.

Recommendation: Use cautious phrasing like “consistent with potential improvements” without implying causality.

Response: We agree. We removed causal phrasing such as "improvement" or "survival gains" from the Results and Conclusion. The revised Discussion states that declining rates may reflect model assumptions, demographic changes, background mortality changes, covariate patterns, and regional information sharing in addition to any true changes in survival.

Comment: Sex differences overinterpretation

3. Sex Differences overinterpretation--The manuscript claims excess male mortality, but the wide UI (0.47–2.33) includes the possibility of no effect or the opposite.

The conclusion lacks statistical support.

Recommendation: Tone down language- Replace “suggested excess male mortality” with: “point estimates indicate possible differences, but uncertainty is substantial”

Response: We revised the Abstract, Results, and Discussion to state that sex differences are uncertain and hypothesis-generating. The Abstract now states: "Point estimates suggested possible higher male mortality, but uncertainty intervals were wide and compatible with no clear sex difference".

Comment: Limited engagement with regional literature

4. Limited Engagement with Existing Literature--The introduction is strong, but the discussion has inadequate context:

Missing comparison with other West African countries, previous GBD SCD studies, benchmarking against Nigeria, Ghana, etc.

Recommendation:

Include regional comparisons and address how Sierra Leone's trends relate to others—whether typical or unique.

Response: We expanded the Discussion to compare the Sierra Leone estimates with global and regional literature, including the GBD 2021 Sickle Cell Disease Collaborators, Williams' review of sub-Saharan Africa, CONSA, and Jamaican newborn screening and cohort evidence. We frame these comparisons cautiously and do not infer that Sierra Leone has achieved similar intervention effects.

REVIEWER #2

1. Is the manuscript technically sound, and do the data support the conclusions?

The authors produced a sound document with data from the Global Burden of Disease (GBD) 2023 that was pre-existing and available in the public domain. All analytical codes utilised for data extraction, and analyses was readily available. The data provided supports the conclusions that were drawn.

2. Has the statistical analysis been performed appropriately and rigorously?

The statistical analyses, though a secondary analyses, was appropriate for the nature of information that the authors wanted to derive. The analysis rightfully encompasses the sickle cell haemoglobinopathies that carry significant morbidity and mortality that include sickle cell anemia (HbSS), sickle-hemoglobin C disease (HbSC), and sickle-beta-thalassemia (HbS/β-thal). Absolute counts of prevalence, deaths, years lived with disability (YLDs), years of life lost (YLLs), and disability-adjusted life-years (DALYs) were recorded. The estimated annual percentage change in age-standardized rates was documented with p-values and confidence intervals. Age and sex stratification was reasonable, and temporal trends were appropriately assessed using log-linear regression. The Kitagawa-Das Gupta decomposition was also an appropriate method to compare mortality trends. The limitations of assumptions made were highlighted in the discussion and some sections of the methodology.

Tables and diagrams were formatted and labelled appropriately.

3. Have the authors made all data underlying the findings in their manuscript fully available?

The authors have made all data, codes utilised for data extraction and analyses readily available on an online platform - GitHub.

4. Is the manuscript presented in an intelligible fashion and written in standard English?

Yes, mostly. Minor corrections highlighted below:

Comment: Rationale and policy impact

Introduction

The authors describe the elements of the introduction relatively well. The background knowledge of Sickle cell Disease was appropriately mentioned, followed by the current knowledge and the gaps in knowledge. The objectives of this study were also highlighted clearly. The authors have not quite elucidated clearly the rationale for partaking in this study. What will be the impact of this intervention? That is not apparent in the introduction. Is it to change policy for instance?

Response: Thank you. We added an Introduction paragraph explaining why a country-specific modeled burden analysis can be useful for policy where surveillance is weak, while also noting its limitations.

Comment: Methods should remain factual

Methodology

The methodology section should present a clear and factual description of the methods used in the study. Discussion of the limitations of these methods is not appropriate in this section and should instead be addressed in the discussion section.

Response: We moved methodological caveats and interpretive limitations out of the Methods where possible and into the Strengths and limitations subsection.

Comment: Other contributors to health shocks

Conclusions

The authors may need to consider that there is more that may contribute to the ‘health shocks.’ Consider the additive effect of the HIV epidemic, malnutrition, the effect of malaria as well as the effect of the covid 19 outbreak on infant mortality rate in that region during that period.

Response: We added cautious language noting that background child mortality and service-use changes related to malaria, malnutrition, HIV, Ebola, and COVID-19 may influence modeled trends and cannot be separated in this descriptive analysis.

REVIEWER #3

Comment: Very good paper. The following changes need to be made:

1) abstract section: Objectives of the study need to be clearly stipulated. Separate results from conclusions. Let the conclusion speak to the title, objectives and the results derived from the study which was global burden of disease trends

Response: We rewrote the Abstract with the requested subsections: Background, Objective, Methods, Results, and Conclusions. The objective is now explicit, and the conclusion is separated from the numeric results.

Comment:

2) Give the global trends of SCD, then regional then local in the introduction section.

Response: We reorganized the Introduction so it moves from global burden, to sub-Saharan Africa and West Africa, to Sierra Leone's evidence gap and health-system context, then to GBD's usefulness and limitations, and finally to the study objective.

Comment: Historical data collection and diagnostic testing

3) Which data collection methods were used in 1990 and how well kept were they? Was the collection standardised back then or can this then be mentioned as a limitation of the study being that it was historical data?

4) In 1990 what tests were being used to diagnose Sickle Cell Disease for confirmatory diagnosis? This needs to be mentioned and what tests are being used now for the same? Were they screening tests or diagnostic tests?

Response: Because this study uses GBD modeled estimates rather than primary data collection, we cannot describe Sierra Leone-specific diagnostic tests used in 1990 or across the full period. We clarified this in the Methods and expanded the limitations to state that sparse historical primary data, underdiagnosis, and possible mismatch between GBD categories and local clinical phenotypes limit interpretation.

Comment: Insert actual figures

5)Pages 10,11,12 insert the actual figures indicated

Response: We removed all figure placeholder text and inserted complete captions and embedded figure versions in the manuscript. High-resolution figure files are provided separately in tiff formats.

Comment: Compare similar studies and avoid recommendations in Results

6) In the discussion , please compare similar studies done for these trends that may have employed the Kitagawa-Das Gupta Decomposition to explain the findings. Refrain from making recommendations in the discussion section, these can be done after the conclusion

Response: We kept Results descriptive and moved interpretation to the Discussion. We expanded the discussion of GBD and regional sickle cell literature, while framing recommendations under a separate "Implications for Sierra Leone" subsection / paragraphs.

Comment: Separate Results from Discussion

7) Separate the results from the discussion, refrain from explaining the results in the results section, this should be done in the discussion section with relevant studies referenced.

Response: We revised the Results section to describe estimates without attributing causes or making policy interpretations. Interpretive material now appears in the Discussion.

Comment: Distinct limitations section

8)The limitations of the study need to be in their own section

Response: We added a clearly labeled Strengths and limitations subsection with detailed limitations specific to GBD modeled estimates, derived age groups, sex ratios, denominators, genotype specificity, and causal inference.

Comment: Conclusion clarity and concision

9) Conclusion is not clear and concise. Please make it better in keeping with the title and objectives of the study

10) Let the information flow from one section to the other in keeping with the study title and don't forget to add the objectives as well.

Response: We rewrote the conclusion to align with the objective and results and to use cautious language about modeled estimates.

REVIEWER #4

Comment:

Overall, this is a well-analyzed paper that presents important population-based information regarding SCD in Sierra Leone. Only minor revisions are suggested, including ensuring that the figures are inserted in the appropriate spaces in the paper. Additionally, I would suggest re-examining the paper to see how certain sections could be made more concise. The discussion is quite lengthy and includes speculative statements that may be reduced or consolidated to streamline the paper more.

Response: Thank you for the supportive assessment. We inserted figure captions in the correct locations, removed placeholders, redid clearer versions of Figures 1, 2, and 4, and consolidated speculative discussion material to improve concision.

REVIEWER #5

In their manuscript Jalloh et al elegantly reuse data from a previous study to provide in depth analysis od sickle cell disease burder in Sierra leone. Such data are rare and deserve publication, however numerous technical and formal points need to be adresses prior to improve the overall value of the manuscript, here are some examples:

Comment: Abstract results clarity

- the results paragraph of the abstract does not protray clearly the content of the paper.

Response: We rewrote the Abstract Results subsection to present the main numeric findings more clearly and separately from the conclusion.

Comment: Prevalence terminology

- prevalence is in %, here the authors present absolute value. Personally, i encourage to express the data as % of the population to improve readibility.

Response: We clarified that prevalence is reported as GBD-estimated prevalent case counts and rates per 100,000 population. We retained absolute counts because they are central to service planning and are paired with rate estimates in Table 1.

Comment: Definition of sickle cell disorders

"sickle cell disaese shoud be defined at the beginning of the manuscript: SS only SC other componds or phenotypically defines?

Response: We now define the GBD sickle cell disorders cause category early in the Introduction and again in Methods, including HbSS, HbSC, and HbS/beta-thalassemia as captured by GBD.

Comment: Sex-based differences

- the sex base difference is poorly explained and deserved a bit more effort.

Response: We revised the sex-difference language throughout to avoid overinterpretation and to emphasize wide uncertainty intervals.

Comment: Statistics and scientific English

In addition to overall statistics should be doubleproofed as well as the English scientific language reviewed.

Response: We checked the tables against the analytic outputs, figures for clarity, and edited the manuscript for grammar, consistency, and cautious scientific tone.

REVIEWER #6

I would li

---

## [Decision Letter · Decision Letter 1]

21 May 2026

Trends in the burden of sickle cell disorders in Sierra Leone, 1990–2023: an analysis of Global Burden of Disease Study 2023 estimates

PONE-D-26-09105R1

Dear Dr. Jalloh,

We’re pleased to inform you that your manuscript has been judged scientifically suitable for publication and will be formally accepted for publication once it meets all outstanding technical requirements.

Kind regards,

Mehmet Baysal

Academic Editor

PLOS One

Additional Editor Comments (optional):

Reviewers' comments:

Reviewer's Responses to Questions

**Comments to the Author**

1. If the authors have adequately addressed your comments raised in a previous round of review and you feel that this manuscript is now acceptable for publication, you may indicate that here to bypass the “Comments to the Author” section, enter your conflict of interest statement in the “Confidential to Editor” section, and submit your "Accept" recommendation.

Reviewer #1: All comments have been addressed

Reviewer #2: All comments have been addressed

Reviewer #3: (No Response)

Reviewer #5: All comments have been addressed

2. Is the manuscript technically sound, and do the data support the conclusions?

Reviewer #1: Yes

Reviewer #2: Yes

Reviewer #3: Yes

Reviewer #5: Yes

3. Has the statistical analysis been performed appropriately and rigorously? 

Reviewer #1: I Don't Know

Reviewer #2: Yes

Reviewer #3: Yes

Reviewer #5: Yes

4. Have the authors made all data underlying the findings in their manuscript fully available?

Reviewer #1: Yes

Reviewer #2: Yes

Reviewer #3: Yes

Reviewer #5: Yes

5. Is the manuscript presented in an intelligible fashion and written in standard English?

Reviewer #1: Yes

Reviewer #2: Yes

Reviewer #3: Yes

Reviewer #5: Yes

6. Review Comments to the Author

Reviewer #1: (No Response)

Reviewer #2: The authors have addressed all comments, including those of other authors appropriately. Specifically, the following concerns were addressed appropriately.

1. Rationale for the study and policy impact

2.Methods section should remain factual

3.Other contributors to health shocks in the conclusion section.

Reviewer #3: Markedly improved. However there are some things that need to be in the final manuscript for it to read bette:

1. Abstract is too short and vague. Add more data to the background. Use a summary from the introduction.

2. The objectives need to be included in the main text as well

3. There are figures in text but no actual figures in the manuscript

Reviewer #5: Thank you for adressing all comments to satisfactory levels. I have no further remakrs at this point.

7. PLOS authors have the option to publish the peer review history of their article (what does this mean?). If published, this will include your full peer review and any attached files.

Reviewer #1: No

Reviewer #2: No

Reviewer #3: No

Reviewer #5: No

---

## [Editor Report · Acceptance letter]

PONE-D-26-09105R1

PLOS One

Dear Dr. Jalloh,

I'm pleased to inform you that your manuscript has been deemed suitable for publication in PLOS One. Congratulations! Your manuscript is now being handed over to our production team.

Kind regards,

on behalf of

Dr. Mehmet Baysal

Academic Editor

PLOS One